# Digital Eddy Current Detection Method Based on High-Speed Sampling with STM32

**DOI:** 10.3390/mi15060775

**Published:** 2024-06-11

**Authors:** Xiong Cao, Erlong Li, Zilan Yuan, Kaituo Zheng

**Affiliations:** School of Mechanical Engineering, Sichuan University, Chengdu 610065, China; caoxiong199@126.com (X.C.); yuanzilan@stu.scu.edu.cn (Z.Y.); 18184619371@163.com (K.Z.)

**Keywords:** eddy current detection, high-speed sampling, digital demodulation, sinusoidal fitting, STM32 microcontroller

## Abstract

The electromagnetic eddy current non-destructive testing system enables the non-destructive analysis of surface defect information on tested materials. Based on the principles of eddy current detection, this paper presents a digital eddy current detection method using high-speed sampling based on STM32. A differential eddy current coil is used as the detection probe, and the combination of a differential bridge and a differential amplifier circuit helps to reduce common-mode noise interference. The detection signal is collected via an STM32-based acquisition circuit and transmitted to the host computer through Ethernet for digital demodulation processing. The host computer performs operations such as smoothing averaging, sinusoidal fitting, and outlier removal to extract the amplitude and phase of the detection signal. The system also visually displays the condition of the tested object’s surface in real time through graphical visualization. Testing showed that this system can operate at frequencies up to 8.84 MHz and clearly identify defects as narrow as 1 mm on the surface of the tested steel plate.

## 1. Introduction

Eddy current testing (ECT) is a classical method of non-destructive evaluation. ECT relies on the interaction between a primary magnetic field and the test material, inducing eddy currents within the object. These currents can reveal discontinuities such as cracks, corrosion, and material degradation, which are detected by monitoring changes in coil impedance or measuring the induced magnetic field [1]. ECT offers a high level of sensitivity for identifying materials and assessing microstructural states [2]. Due to its high sensitivity, straightforward design, and non-invasive nature—requiring no coupling agents and not compromising the integrity of the conductor’s internal structure—ECT is extensively applied across various fields. Notable applications include Raude et al.’s use of eddy current arrays for detecting stress corrosion cracking [3], Bailey et al.’s study employing GMR sensor arrays to characterize corrosion beneath coatings on pipelines [4], Grosso et al.’s application of multifrequency eddy currents and signal processing to identify ferric oxide in petrochemical storage tanks [5], and M. Grosso et al.’s utilization of both eddy currents and ultrasonic techniques for the non-destructive detection of in-plane wrinkles in carbon fiber laminates [6]. These studies have all yielded significant detection results.

Eddy current testing (ECT) is based on the principles of electromagnetic induction and the eddy current effect, primarily involving two components: coil excitation to generate an eddy current field and probe measurement. When an alternating current is applied to an excitation coil, it generates an alternating magnetic field around the coil according to Faraday’s law of electromagnetic induction. As the coil approaches the test object, the magnetic field induces eddy currents within the object. These eddy currents intersect with the excitation magnetic field, forming a time-varying magnetic field that impedes changes in the excitation field [7]. This interaction induces detectable voltage variations in the detection probe’s output impedance. The presence of defects in the conductor distorts the eddy currents, altering the magnetic field, which is reflected in changes in the amplitude and phase of the probe’s output voltage. By analyzing these changes, information related to the damage can be deduced. Traditionally, ECT utilized coil probes, with the excitation coil serving as the detection sensor, where the eddy currents affect the excitation coil’s impedance changes. Subsequently, researchers discovered that magnetic sensors (Hall elements or magnetoresistive sensors) could replace coil probes and directly respond to changes in eddy currents [8,9]. Modern classical ECT systems typically consist of an eddy current field excitation device, an eddy current detection device, a signal processing unit, a data acquisition system, and a data analysis system. The eddy current field excitation device often uses direct digital synthesis (DDS) technology, known for its flexibility, rapid frequency changes, and simplicity. However, this technology is costly and has limited interference resistance. The detection device mainly consists of excitation coils and detection probes. Coil probes are simple in structure but have poor interference resistance and directivity; magnetic probes offer higher sensitivity and interference resistance but demand high linearity from the magnetic sensors, increasing their size and cost. Differential coil probes [10,11] based on a differential Wheatstone bridge not only provide high sensitivity and linearity but also are cost-effective and compact, making them widely used in ECT setups. The signal processing unit typically includes instrument amplifiers, band-pass filters, and orthogonal-vector lock-in amplifiers. The defect signals output by the ECT device are amplified by instrument amplifiers, with high- and low-frequency interference filtered out by band-pass filters, followed by orthogonal-vector demodulation to detect the amplitude and phase information of defect signals [12]. This circuit design facilitates the analog detection process, but it also complicates the structural design and increases costs. Data acquisition and analysis devices often employ an FPGA (Field-Programmable Gate Array) [13,14,15] or DSP (Digital Signal Processor) [16,17], which sends processed signals to a host computer via a serial port for real-time display. Reference [18] describes a digital signal processing system specifically designed for eddy current detection. This system is built on an FPGA and a DSP processing core featuring high-speed data conversion capabilities and can be flexibly configured with various peripherals to meet different detection requirements. Similarly, the eddy current detection system in reference [19] is also based on an FPGA, uses direct digital synthesis technology to generate excitation signals, and employs in-phase and quadrature demodulation techniques to calculate the real and imaginary parts of the probe output signal, thereby obtaining information about the test object. However, FPGAs are expensive and complex, involving intricate hardware logic design, timing constraints, and the optimization of wiring. DSPs require specialized knowledge of data processing and optimization, making development and debugging challenging. The use of serial ports to receive data on the host computer limits the speed and distance of data transmission. Compared to the FPGA and DSP, the STM32 microcontroller, due to its high level of integration, can reduce the need for external components, thereby lowering costs and making it suitable for large-scale deployment and applications with limited budgets. Additionally, the STM32 benefits from extensive development tools and community support and involves less complex timing logic and specialized knowledge, making its development easier. Consequently, STM32 microcontrollers are widely used in portable devices, the Internet of Things, and simple control applications.

The key to eddy current testing technology lies in measuring the amplitude and phase information of defect signals. This paper introduces a digital eddy current detection method based on high-speed data acquisition with an STM32 microcontroller, achieving high precision in a differential eddy current non-destructive testing sensor system design. The method uses instrument amplifiers to boost defect signals, which are then captured by the STM32 and transmitted to a host computer via Ethernet. The host computer employs MATLAB R2023a software to digitally demodulate the defect data using a three-parameter sinusoidal fitting algorithm [20,21,22], extracting both the amplitude and phase information of the defect signals and analyzing the fitting accuracy. This digital demodulation method sidesteps the complexities associated with traditional analog circuitry and FPGA hardware logic design. It features a simpler architecture, lower costs, reduced power consumption, and greater ease of operation. Ethernet’s capacity for long-range data transmission makes it ideal for applications requiring rapid data collection and transmission in eddy current testing scenarios. This paper details the process of implementing high-speed data acquisition with STM32 and using host computer MATLAB software for demodulation to detect the amplitude and phase information of defect signals. Finally, this paper discusses the construction of an experimental eddy current testing setup and presents the experimental results.

## 2. Principle of Eddy Current Testing

In eddy current testing, the impedance parameter ${\dot{Z}}_{0}$ for each coil is a complex number, defined as in Equation (Equation 1), which represents the ratio of the voltage to the current $\left(V_{0}/I_{0}\right)$ under a single-frequency harmonic excitation *f* [1]: (1)$${\dot{Z}}_{0}=\frac{V_{0}}{I_{0}}=R_{0}+jX_{0}$$

Here, *j* represents the imaginary unit. The real part of the complex impedance parameter ${\dot{Z}}_{0}$ is the coil’s resistance *R*, and the imaginary part is the coil’s inductive reactance *L*. According to Maxwell’s electromagnetic theory, when a harmonic excitation of frequency *f* passes through the coil, the coil’s inductive reactance is related to the frequency *f*. This relationship can be expressed as shown in Equation (Equation 2): (2)$$X_{0}=\omega L_{0}=2\pi fL_{0}$$

Here, ω is the angular frequency, *f* is the frequency of the harmonic excitation, and $L_{0}$ is the coil inductance. The coil’s impedance parameter ${\dot{Z}}_{0}$ is a complex number. According to the definition of complex numbers, the impedance can be expressed using its magnitude $|{\dot{Z}}_{0}|$ and phase φ, where the magnitude $|{\dot{Z}}_{0}|$ is calculated as the square root of the sum of the squares of the real and imaginary parts ($|{\dot{Z}}_{0}|=\sqrt{R_{0}^{2}+X_{0}^{2}}$), and the phase φ is determined by the arctangent of the ratio of the imaginary part to the real part ($\varphi =\arctan\left(X_{0}/R_{0}\right)$). Following Equations (1) and (2) and the principles of complex numbers, the complex impedance parameter ${\dot{Z}}_{0}$ can be further described as given in Equation (Equation 3): (3)$${\dot{Z}}_{0}=\frac{V_{0}}{I_{0}}=R_{0}+jX_{0}=R_{0}+j2\pi fL_{0}={\sqrt{R_{0}^{2}+X_{0}^{2}}}_{\varphi =\arctan\left(X_{0}/R_{0}\right)}={\left|\dot{Z_{0}}\right|}_{\varphi }$$

When a metal conductor is placed within a changing magnetic field, it generates induced currents that flow through the conductor in patterns similar to water whirlpools, known as eddy currents. This phenomenon is known as the eddy current effect. Figure 1 illustrates the phenomenon where an induced current is generated in the conductor due to the changing magnetic flux density when an alternating current is passed through the primary coil, as defined by Faraday’s law of electromagnetic induction in Equation (Equation 4). The electromotive force (ε) is directly proportional to the rate of change in the magnetic flux density over time, represented by $\Phi_{B}$ [1].
(4)$$\varepsilon =-\frac{d\Phi_{B}}{dt}$$

When the excitation coil is brought close to the test object, the object induces circular currents (eddy currents) under the influence of an alternating magnetic field. These eddy currents generate a secondary magnetic field that acts back on the original coil. This reverse magnetic field from the conductive material weakens the primary magnetic field, resulting in changes in the coil’s impedance [7,23]. As shown in Figure 2, when the test object is free of defects, the eddy currents flow in patterns similar to water whirlpools; however, when defects are present in the test object, the flow path of the eddy currents is altered. This alteration affects the secondary magnetic field generated by the eddy currents, which, in turn, changes the impact on the coil’s impedance. By using sensors to detect changes in the coil’s impedance, defects in the test object can be assessed.

Figure 3 illustrates the coil–test object interaction based on the transformer model: ${\dot{U}}_{o}$ represents the harmonic excitation at angular frequency ω; $R_{o}$ represents the impedance of the excitation coil; $j\omega L_{o}$ denotes the inductive reactance of the excitation coil; $R_{e}$ represents the impedance of the test object; $j\omega L_{e}$ denotes the inductive reactance of the test object; and *M* represents the mutual inductance coefficient between the excitation coil and the test object. Based on the principles of magnetic coupling and mutual inductance between coils, the current ${\dot{I}}_{o}$ in the excitation coil generates an equivalent current of magnitude $-j\omega M{\dot{I}}_{o}$ on the test object (in the same direction as ${\dot{I}}_{e}$), while the induced current on the surface of the test object, in turn, affects the excitation coil, producing an equivalent current effect $-j\omega M{\dot{I}}_{e}$ (in the same direction as ${\dot{I}}_{o}$).

According to Kirchhoff’s laws, Equation (Equation 5) is established: (5)$$\left\{\begin{array}{c} \left(R_{o}+j\omega L_{o}\right){\dot{I}}_{o}-j\omega M{\dot{I}}_{e}={\dot{U}}_{o}\\ \left(R_{e}+j\omega L_{e}\right){\dot{I}}_{e}-j\omega M{\dot{I}}_{o}=0 \end{array}\right.$$

Solving the equation system in Equation (Equation 5) yields the expressions for the excitation coil current ${\dot{I}}_{o}$ and the induced current ${\dot{I}}_{e}$, as defined in Equation (Equation 6), where all variables have the meanings depicted in Figure 3.
(6)$$\left\{\begin{array}{c} {\dot{I}}_{o}=\frac{{\dot{U}}_{o}}{R_{o}+\frac{\omega^{2}M^{2}}{R_{e}^{2}+{\left(\omega L_{e}\right)}^{2}}R_{e}+j\left[\omega L_{o}-\frac{\omega^{2}M^{2}}{R_{e}^{2}+R^{2}+{\left(\omega L_{e}\right)}^{2}}\omega L_{e}\right]}\\ {\dot{I}}_{e}=\frac{\omega^{2}ML_{e}+j\omega MR_{e}}{R_{e}^{2}+{\left(\omega L_{e}\right)}^{2}}{\dot{I}}_{o} \end{array}\right.$$

Using Equations (5) and (6), we can express the equivalent resistance and inductance of the primary coil, which are influenced by the eddy current magnetic field generated by the test object, as detailed in Equation (Equation 7): (7)$$\left\{\begin{array}{c} R=R_{o}+R_{e}\omega^{2}M^{2}/\left(R_{e}^{2}+\omega^{2}L_{e}^{2}\right)\\ L=L_{o}-L_{e}\omega^{2}M^{2}/\left(R_{e}^{2}+\omega^{2}L_{e}^{2}\right) \end{array}\right.$$

The formulas for equivalent resistance *R* and inductance *L* show that the changes in these parameters for the excitation coil after being influenced by the test object are related not only to its own resistance $R_{o}$ and inductance $L_{o}$ but also to the angular frequency ω of the excitation signal, the mutual inductance coefficient *M* between the test object and the excitation coil, and the resistance $R_{e}$ and inductance $L_{e}$ of the test object. The presence of defects in the test object will cause changes in $R_{e}$ and $L_{e}$, which, in turn, affect the equivalent resistance and inductance of the excitation coil. Detecting these changes with a sensor can provide information about surface defects on the test object. A common eddy current detection method involves measuring the amplitude and phase of the output signal after it passes through the excitation coil, specifically involving the analysis of the signal amplitude and phase at test point 1, as depicted in Figure 3. This paper presents a differential coil probe based on a differential bridge as the eddy current detection probe, which is discussed in Section 3.

## 3. Components of an Eddy Current Testing System

### 3.1. System Components

The electromagnetic eddy current non-destructive testing system designed in this paper primarily consists of several components: a signal generator, sensor probe (excitation coil), AC bridge circuit, signal amplification circuit, AD9226 acquisition circuit, STM32F407, Ethernet interface circuit, and host computer. Figure 4 presents the system architecture diagram. Here, one channel of the signal generator produces a harmonic excitation that supplies the excitation signal to the coil, inducing eddy currents in the test conductor. A square-wave signal, synchronized in frequency and phase with the harmonic excitation, is fed into the STM32 main controller to serve as a reference for signal acquisition and provide a basis for subsequent phase computation. The test signal containing defect information, the output from the AC bridge, is then amplified by the signal amplification circuit. The amplified defect signal is captured by the AD9226 acquisition board and transmitted to the host computer via the STM32 and Ethernet, where the host computer digitally demodulates the defect signal, performs amplitude and phase analysis, and visually displays the data. Section 5 details the demodulation process of the amplitude and phase.

The system design eliminates the need for an analog demodulation circuit by employing host computer programming to digitally demodulate the amplified defect signals.

### 3.2. AC Bridge

This paper utilizes a differential coil probe based on a differential bridge circuit for eddy current detection, with Figure 5 illustrating the AC bridge circuit. ${\dot{U}}_{i1}$ is the negative input to the amplifier, and ${\dot{U}}_{i2}$ is the positive input. The coils $L_{1}$ and $L_{2}$ serve as differential coils, their function being to detect changes in the eddy currents and convert them into changes in their own impedance. The output voltage $\dot{U_{o}^{\prime }}$ of the AC bridge can be represented by Equation (Equation 8): (8)$$\left\{\begin{array}{c} \dot{U_{0}^{\prime }}={\dot{U}}_{i1}-{\dot{U}}_{i2}=\frac{{\dot{Z}}_{1}{\dot{Z}}_{4}-{\dot{Z}}_{2}{\dot{Z}}_{3}}{\left({\dot{Z}}_{1}+{\dot{Z}}_{2}\right)\left({\dot{Z}}_{3}+{\dot{Z}}_{4}\right)}\\ {\dot{Z}}_{1}=R_{1},{\dot{Z}}_{2}=R_{2},{\dot{Z}}_{3}=j\omega L_{1},{\dot{Z}}_{4}=j\omega L_{2} \end{array}\right.$$

The balance condition of the bridge circuit is given by Equation (Equation 9): (9)$${\dot{Z}}_{1}{\dot{Z}}_{4}-{\dot{Z}}_{2}{\dot{Z}}_{3}=0$$

When the harmonic excitation passes through the differential coils, it causes the impedance of the bridge to change due to defect information on the surface of the test object, resulting in an unbalanced condition that outputs a differential signal. Considering that the bridge output signal is small and may contain substantial common-mode noise, it is necessary to amplify the bridge output signal using an instrumentation amplifier before it can be acquired by the AD converter.

According to the differential amplifier circuit equations [24,25], when $\frac{R_{f}}{R_{3}}=\frac{R_{p}}{R_{4}}$, the output of the amplifier is ${\dot{U}}_{o}=\frac{R_{f}}{R_{3}}\left({\dot{U}}_{i2}-{\dot{U}}_{i1}\right)$ (where ${\dot{U}}_{i1}$ is the input at the negative terminal, and ${\dot{U}}_{i2}$ is the input at the positive terminal), at which point the common-mode rejection ratio of the amplifier approaches infinity. The advantage of using an AC differential bridge lies not only in doubling the sensitivity of the system but also in providing temperature compensation.

## 4. Software Design for the STM32 Microcontroller

The eddy current detection system of the lower computer described in this paper is primarily controlled through Keil5 programming of the STM32. The main functionalities implemented include the high-speed acquisition of defect signals and high-speed Ethernet data transmission between the lower and host computers. The program flowchart for the lower computer is shown in Figure 6.

High-speed data acquisition in the STM32 is facilitated through the FSMC bus, enabling the rapid parallel transmission of 12-bit data captured by the AD9226. The basic process of data acquisition includes capturing an interrupt at the rising edge of a reference square wave; when the interrupt occurs, a flag is set. The main loop then checks whether the flag is set to start data acquisition from the AD chip and writes the read data into the data buffer. Once the buffer is full, the flag is cleared to provide a baseline for the phase calculation of the detection signal. Communication between the upper and lower computers is implemented via ETH. In the STM32 program, the Lwip protocol stack is ported to conveniently enable Ethernet access. The basic process for Ethernet data transmission in the STM32 involves activating the DHCP function to automatically acquire an IP address (or using a static IP if acquisition fails), allowing the user to set the remote IP address according to the host computer’s IP address. After completion, the program remains in monitoring mode until the data reception buffer is full, at which point it allocates memory for the pbuf structure, writes the buffered data into this structure to form a linked list, and then sends out the data. This system is designed for high-speed data transmission, meeting the requirements of eddy current detection.

## 5. Digital Demodulation by the Host Computer

### 5.1. Overview of Host Computer Functions

In the digital demodulation system described in this paper, a host computer program written in MATLAB is used to receive, demodulate, and visually display defect signals. The design of the MATLAB program must consider two aspects: firstly, the program should accurately reflect the defects on the surface of the test object, meaning that digital demodulation must provide sufficient amplitude and phase angle accuracy and sensitivity; secondly, the system’s real-time capability is crucial, as it ensures that defects on the surface of the test object are detected in real time, thereby preventing any signal blockages. To meet these requirements, the digital demodulation algorithm must demodulate at a high rate to prevent data blockages while maintaining accuracy.

This paper describes the creation of a UDP port and data processing program in MATLAB to manage data transmitted from the lower computer to the host computer via the Ethernet UDP protocol, as illustrated in Figure 7. The program’s basic functions and main processes include receiving and converting data sent from the lower computer and eliminating outlier signals at the edges. The converted result is the detection signal containing defect information. The detection signal undergoes low-pass filtering through a moving-average process, and the smoothed signal is then analyzed using a three-parameter least-squares sinusoidal fitting algorithm to extract amplitude and phase information. Finally, the host computer visualizes the transformed amplitude and phase information to facilitate a more intuitive analysis of the defects on the surface of the test object.

### 5.2. Three-Parameter Sinusoidal Fitting Algorithm

The purpose of digitally demodulating eddy current detection signals is to extract amplitude and phase information from the detection signals. This paper employs a three-parameter sinusoidal fitting algorithm [20,21,22] for signal filtering and digital demodulation. The advantage of using this algorithm lies in its closed-loop nature when the frequency is known, ensuring there are no convergence issues. Additionally, it provides effective fitting, operates at high speeds, and prevents data blockages.

When a sinusoidal excitation signal with frequency *f* is input into the signal input end of the eddy current system, the form of the signal output by the eddy current coil and detection device is described by Equation (Equation 10): (10)$$V_{i}\left(t\right)=A\sin\left(2\pi ft+\varphi \right)+C$$

Here, *A* represents the amplitude of the detection signal; *f* is the frequency of the output signal, which remains consistent with the frequency of the input signal and is a known quantity; φ is the phase angle of the output signal; and *C* is the DC offset of the output signal, which, theoretically, is very small. For the eddy current detection system, the useful characteristic signals that need to be extracted and analyzed are the signal amplitude *A* and the output phase angle φ.

Equation (Equation 10) is transformed into Equation (Equation 11) through trigonometric transformation, where a=Asin(φ)andb=Acos(φ). By fitting the parameters *a*, *b*, and *C*, the expressions for the amplitude and phase of the detection signal can be obtained, as shown in Equation (Equation 12): (11)V(t)=acos(2πft)+bsin(2πft)+C
(12)$$\left\{\begin{array}{c} A\left(a,b\right)=\sqrt{a^{2}+b^{2}}\\ \Phi \left(a,b\right)=\left\{\begin{array}{c} \tan^{-1}\left(\frac{a}{b}\right);\;b\geq 0\\ \tan^{-1}\left(\frac{a}{b}\right)+\pi ;\;b<0 \end{array}\right. \end{array}\right.$$

For digital sampling signals, where the number of samples per cycle is *n*, the frequency of the digital signal is 1/n. The digital signal can be represented by Equation (Equation 13): (13)$$V_{i}=a\cos\left(\left(2\pi /n\right)\times i\right)+b\sin\left(\left(2\pi /n\right)\times i\right)+C$$

The underlying principle is to minimize the sum of squared residuals in the process of three-parameter fitting for digital sinusoidal signals. The sum of squared residuals between the sampled data and the results of the function fit is expressed by Equation (Equation 14): (14)$$\xi \left(a,b,C\right)=\sum_{i=1}^{n}{\left[V_{i}^{\prime }-\left(a\cos\left(\frac{2\pi }{n}i\right)+b\sin\left(\frac{2\pi }{n}i\right)+C\right)\right]}^{2}$$

Here, $V_{i}^{\prime }$ represents the true value for each sample.

To minimize the sum-of-squared fitting errors, the condition that must be met is given by Equation (Equation 15): (15)$$\frac{\partial \xi }{\partial a}=0;\;\frac{\partial \xi }{\partial b}=0;\;\frac{\partial \xi }{\partial c}=0$$

If we set: (16)$$\alpha_{i}=\cos\left(\frac{2\pi }{n}i\right);\beta_{i}=\sin\left(\frac{2\pi }{n}i\right)$$
then Equation (Equation 15) can be expanded to be expressed as Equation (Equation 17): (17)$$\left\{\begin{array}{c} \sum_{i=1}^{n}\alpha_{i}^{2}\times a+\sum_{i=1}^{n}\alpha_{i}\beta_{i}\times b+\sum_{i=1}^{n}\alpha_{i}\times C=\sum_{i=1}^{n}V_{i}^{\prime }\cdot \alpha_{i}\\ \sum_{i=1}^{n}\alpha_{i}\beta_{i}\times a+\sum_{i=1}^{n}\beta_{i}^{2}\times b+\sum_{i=1}^{n}\beta_{i}\times C=\sum_{i=1}^{n}V_{i}^{\prime }\cdot \beta_{i}\\ \sum_{i=1}^{n}\alpha_{i}\times a+\sum_{i=1}^{n}\beta_{i}\times b+\sum_{i=1}^{n}1\times C=\sum_{i=1}^{n}V_{i}^{\prime } \end{array}\right.$$

By solving the equation system in Equation (Equation 17), the optimal fit values for *a*, *b*, and *C* can be determined. This system can be solved in matrix form. The equation system in Equation (Equation 17) can be represented as the product of matrix *B* and matrix *S*, as shown in Equation (Equation 18):(18)B×S=Y

The matrices *B*, *S*, and *Y* can be represented by Equation (Equation 19): (19)$$B=\left[\begin{array}{ccc} \sum_{i=1}^{n}\alpha_{i}^{2} & \sum_{i=1}^{n}\alpha_{i}\beta_{i} & \sum_{i=1}^{n}\alpha_{i}\\ \sum_{i=1}^{n}\alpha_{i}\beta_{i} & \sum_{i=1}^{n}\beta_{i}^{2} & \sum_{i=1}^{n}\beta_{i}\\ \sum_{i=1}^{n}\alpha_{i} & \sum_{i=1}^{n}\beta_{i} & \sum_{i=1}^{n}1 \end{array}\right];S=\left[\begin{array}{c} a\\ b\\ C \end{array}\right];Y=\left[\begin{array}{c} \sum_{i=1}^{n}V_{i}^{\prime }\cdot \alpha_{i}\\ \sum_{i=1}^{n}V_{i}^{\prime }\cdot \beta_{i}\\ \sum_{i=1}^{n}V_{i}^{\prime } \end{array}\right]$$

Sinusoidal fitting allows for the determination of the amplitude of the sampled signal. For a standard sinusoidal excitation signal input, the three-parameter sinusoidal fitting implemented in MATLAB is illustrated in Figure 8. The fitting results meet the basic requirements of digital demodulation and enable the extraction of information on the signal’s amplitude and phase angle.

### 5.3. Fit Accuracy Analysis

For the sinusoidal parameters obtained through the three-parameter least-squares method, accuracy considerations include the precision of the detection signal, along with the accuracy of the fitted amplitude and phase. This precision estimation is typically analyzed using the variance or standard deviation. The literature [21,22] indicates that estimates of fitting accuracy can be obtained from the residuals of the fit. The standard deviation of the detection signal is expressed by the sum of squares of the residual errors, as shown in Equation (Equation 20), where ∑v represents the sum of squares of the residual errors, $V_{i}^{\prime }$ represents the sampled signal values (measured values), and *a*, *b*, and *C* are the optimal fit values.
(20)$$\sum v=\sum_{i=1}^{n}{\left[V_{i}^{\prime }-\left(a\cos\left(\frac{2\pi }{n}i\right)+b\sin\left(\frac{2\pi }{n}i\right)+C\right)\right]}^{2}$$

The standard deviation of the detection signal is represented by Equation (Equation 21), where *n* denotes the number of sampling points.
(21)$$\sigma =\sqrt{\frac{\sum v}{n-3}}$$

Regarding the standard deviation of the fitting parameters, the literature [26,27,28] provides an estimation process for the fitting variance. This paper presents the expressions for the standard deviations of $\sigma_{a},\sigma_{b},$ and $\sigma_{C}$, as shown in Equation (Equation 22): (22)$$\sigma_{a}=\sigma \sqrt{d_{11}};\;\sigma_{b}=\sigma \sqrt{d_{22}};\;\sigma_{C}=\sigma \sqrt{d_{33}}$$

The coefficient matrix is represented by Equation (Equation 23), with $\alpha_{i}$ and $\beta_{i}$ having been specifically expressed in Equation (Equation 16). The values $d_{11},d_{22},$ and $d_{33}$ in Equation (Equation 22) are the diagonal values in matrix *M*, which is described by Equation (Equation 24).
(23)$$B=\left[\begin{array}{ccc} \alpha_{1} & \beta_{1} & 1\\ \alpha_{2} & \beta_{2} & 1\\ \vdots & \vdots & \vdots \\ \alpha_{n} & \beta_{n} & 1 \end{array}\right]$$
(24)$$M={\left(B^{T}B\right)}^{-1}=\left[\begin{array}{ccc} d_{11} & d_{12} & d_{13}\\ d_{21} & d_{22} & d_{23}\\ d_{31} & d_{32} & d_{33} \end{array}\right]$$

The standard deviations $\sigma_{A}$ and $\sigma_{\varphi }$ for the fitted amplitude and phase can be synthesized through the function’s standard deviation with the synthesis formula provided in Equation (Equation 25). The specific expressions for A(a,b) and Φ(a,b) in the formula have already been defined in Equation (Equation 12).
(25)$$\left\{\begin{array}{c} \sigma_{A}=\sqrt{{\left(\frac{\partial A(a,b)}{\partial a}\sigma_{a}\right)}^{2}+{\left(\frac{\partial A(a,b)}{\partial b}\sigma_{b}\right)}^{2}}\\ \sigma_{\varphi }=\sqrt{{\left(\frac{\partial \Phi (a,b)}{\partial a}\sigma_{a}\right)}^{2}+{\left(\frac{\partial \Phi (a,b)}{\partial b}\sigma_{b}\right)}^{2}} \end{array}\right.$$

This approach provides estimates of the precision for the detection signal, as well as the fitted amplitude and phase. The reasonableness of the demodulation algorithm can then be assessed through these precision estimates.

### 5.4. Introduction to Other Host Computer Programs

This paper utilizes the read() function provided by MATLAB to retrieve data transmitted by the UDP from the lower computer. The received data are digital quantities converted via an AD converter and need to be transformed into analog quantities that represent voltage values. The conversion formula is presented in Equation (Equation 26), where *V* represents the analog voltage value and data refers to the data collected by the AD converter: (26)$$V=\left(2048-data\right)\times \frac{5}{2048}$$

The output signals from the eddy current detection system are typically weak and subject to significant noise. Therefore, this paper initially applies a preliminary moving average to the detection signals to remove major noise before performing sinusoidal fitting. This approach significantly enhances the fitting effectiveness, enabling the more accurate extraction of amplitude and phase information. For outlier values at the edges of data packets, this study opts to directly exclude them.

Eddy current detection, as a real-time detection system, requires the dynamic visualization of detection signals in real time. Accordingly, this paper employs MATLAB’s plot() and addpoints() functions to provide real-time visual displays of the detection signals, amplitudes, and phases. Through the graphical interface, the variations in detection signals can be monitored live. This is a three-channel display window that allows for the dynamic observation of the test object’s surface information.

## 6. Experimentation and Testing

### 6.1. Introduction to the Test System

To verify the performance of the designed system and its digital demodulation algorithm, this study tests and analyzes a steel plate with identified surface defects. As illustrated in Figure 9, the steel plate has a defect that is 1 mm wide, 250 mm long, and 1 mm deep. The experiment assesses the system’s performance via the real-time dynamic detection of this specific defect.

As shown in Figure 10, the experimental test platform comprises an STM32F407 controller with an AD9226, a detection amplification circuit comprising an AC bridge and differential amplifier, a signal shielding box, a regulated ±5-volt power supply, a router, a signal generator, a host computer, the defective steel plate under test, and an eddy current detection probe.

Comparative experiments reveal that when the sinusoidal excitation signal is at a frequency of 40 KHz and an amplitude of 18 V, the probe exhibits higher sensitivity and lower noise. Therefore, the chosen parameters for sinusoidal excitation are a frequency of 40 KHz and an amplitude of 18 V.

When a sinusoidal excitation signal with a frequency of 40 KHz is directly inputted to the AD acquisition board, the system’s sampling frequency can be determined by performing a four-parameter sinusoidal fitting on the sampled data using the MATLAB toolbox. After multiple tests, the sampling frequency of the system designed in this study was finalized at 8.84 MHz. For conventional eddy current detection systems, the input excitation frequency generally ranges from 10 KHz to 500 KHz. To comply with the Nyquist Sampling Theorem, the system’s sampling frequency must be at least 1 MHz (for an input excitation frequency of 500 KHz). Moreover, to capture more defect information, the sampling frequency should be at least ten times higher than the excitation frequency, i.e., above 5 MHz. The sampling frequency of the system developed in this paper is 8.84 MHz, which meets the requirements for data processing and sampling frequency in eddy current detection systems.

### 6.2. Data Smoothing and Accuracy Analysis

On the one hand, the defect signals output by the eddy current detection system are relatively weak and contain a significant amount of high-frequency noise. On the other hand, the UDP network protocol involved, being an unreliable data transmission protocol, inevitably experiences occasional data loss and single data errors during high-speed data transmission. To minimize the impact of noise and certain random errors on the accuracy of digital demodulation, we will smooth the data and select an appropriate size for the data smoothing window.

The size of the data smoothing window directly affects the precision of the subsequent sinusoidal fitting. To select an appropriate window size, this study conducted comparative experiments using window sizes of 10, 20, 30, 40, 45, and 50 for sinusoidal fitting and analyzed the fitting precision (the fitting methods and precision analysis methods are discussed in Section 5). The results after smoothing and fitting are shown in Figure 11.

Figure 11c illustrates the curves showing how the signal error, amplitude parameter error, and phase angle parameter error from the three-parameter fitting vary with the size of the chosen data smoothing window. The analysis of the fitting errors reveals that increasing the size of the smoothing window gradually reduces the signal error and the errors in amplitude and phase parameters. However, if the smoothing window is too large, the signal loses its sinusoidal characteristics, which leads to increased errors. Therefore, we have chosen a smoothing window size of 40 to achieve the best accuracy. At this size, the standard deviation of the amplitude parameter obtained through three-parameter fitting is about 0.8 mV, and the phase angle standard deviation is about $48^{\prime \prime }$. These values meet the accuracy requirements of the digital demodulation system. Additionally, to obtain more amplitude information without compromising the system’s real-time capabilities, this article employs Matlab’s local maximum finding function, findpeaks(), to extract amplitude data. Upon evaluation, this method was found to provide stable amplitude information that clearly reflects changes in the surface information of the tested material.

It should be noted that, as seen in Figure 11b, the amplitude of the detection signal continuously decreases when we perform data smoothing and the smoothing window size increases. This inevitably introduces errors, resulting in fitting results that do not truly represent the output signals of the eddy current detection system. However, in practical applications of eddy current detection, the system is more concerned with changes in the peak values between defect-free and defective surfaces rather than the magnitude of a specific amplitude. Therefore, the reduction in peak values caused by smoothing operations has a minimal impact on the system, and such effects are negligible and not a primary concern.

### 6.3. Edge Value Elimination

The lower computer system described in this paper sends data in packets, which results in discontinuities at the edges of each data packet. Such discontinuities, serving as the basis for our analysis of the detection signal’s phase, are inevitable.

However, outlier errors may occur at the edges of each packet due to the effects of data smoothing and the initial stability of the system. As shown in the left image of Figure 12, these values undoubtedly impact the phase analysis of the detection signals and therefore need to be eliminated. The approach adopted in this paper is to directly discard the data at the edges of each packet. The right image in Figure 12 shows that the erroneous outlier values at the edges have been effectively removed. The remaining discontinuities at the data packet edges then serve as the reference standard for the phase demodulation of each packet.

It should be noted that the exclusion of data at the edges will introduce a certain error in subsequent phase calculations; however, this error remains constant when the amount of data excluded does not change, constituting a predictable system error that can be compensated for. Furthermore, eddy current detection systems are more concerned with variations in the phase of the detection signal rather than the magnitude of the phase itself. Therefore, the removal of edge values introduces a consistent system error in each phase demodulation, but it does not affect the variations in the signal phase. Thus, it is acceptable to discard erroneous edge data from each packet.

### 6.4. Visual Dynamic Inspection

After applying sliding smoothing, removing edge values, and conducting three-parameter sinusoidal fitting, the digital demodulation system has completed the basic analysis of the detection signal. The next step is to visually display the demodulation results.

As a real-time dynamic detection system, eddy current detection requires real-time updates and visual displays of the amplitude and phase of the detection signal. This paper implements dynamic visual tracking of the data using MATLAB’s plot() and addpoints() functions. When the eddy current probe is stationary on a defect-free surface of the tested steel plate, the detection signal is characterized by the first curve in Figure 13, with the amplitude in the second curve and the phase in the third curve. It is evident that on a defect-free surface, both the amplitude and phase of the detection signal remain stable without significant fluctuations.

When the eddy current probe is moved back and forth over the defect on the steel plate five times, the amplitude of the detection signal is characterized by the second curve in Figure 14, and the phase changes are depicted in the third line. It is observable that both the amplitude and phase exhibit significant fluctuations as the probe passes over the defect five times, with five distinct waveform changes evident in Figure 14. For the defect tested, which is 1 mm wide and 1 mm deep, these changes are pronounced, and the system can dynamically track the movement of the eddy current probe, allowing for real-time updates of measurements. This performance aligns with the requirements of eddy current detection systems.

### 6.5. Discussion

Experimental testing of the system demonstrates that it can clearly respond to surface defects on a steel plate measuring 1 mm in width and 1 mm in depth, as shown in Figure 14, while producing stable outputs when no defects are present, as shown in Figure 13. An analysis of the fitting errors reveals that when a smoothing window size of 40 is selected, the amplitude error of the fitting results is approximately 1 mV, and the phase angle error is about 50 arcseconds (Figure 11c). Such accuracy meets the requirements of eddy current detection systems. The system only needs to amplify the detection signal before directly collecting it. The digital demodulation system then filters the signal and extracts characteristic values, ultimately providing real-time visual tracking displays. This optimizes the system structure, avoiding complex hardware demodulation designs, and achieves a sampling frequency of 8.84 MHz with limited resources.

The challenge in experimentation and testing lies in continually optimizing the STM32 microcontroller program at the lower machine level to eliminate unnecessary operations and interruptions, with the goal of increasing the sampling frequency. Additionally, it is necessary to synchronize the timing of Ethernet data transmission with the refresh rate of the visualization display on the host computer to avoid delays caused by data packet accumulation. This paper implements synchronization between data transmission and visualization refreshing by introducing a reasonable delay after each data packet is sent. Since the system’s sampling occurs within each data packet, delays between packets do not affect the sampling frequency; however, such delays can result in the loss of some defect information when the probe moves too quickly, which is a limitation of the system described in this paper.

## 7. Conclusions

The contribution of this work lies in the design of a high-speed data acquisition and digital demodulation system based on STM32. Compared to traditional eddy current detection systems, our proposed system offers significant advantages. One such advantage is the avoidance of the complex analog demodulation circuit designs traditionally relied upon while still meeting the signal sampling frequency requirements (the designed system’s sampling frequency is 8.84 MHz). Digital demodulation offers flexibility; adjustments to the demodulation program on the host computer can make the system suitable for different eddy current detection signals without the need to tweak analog circuit parameters. Compared to traditional systems that utilize a DSP or an FPGA for digital eddy current detection, the method presented in this paper based on high-speed sampling with the STM32 avoids the complex timing operations and peripheral controls associated with FPGA solutions. Additionally, by utilizing Ethernet for signal transmission, it overcomes the limitations of short-range data transmission inherent in traditional methods. This introduces a new technological approach to the field of industrial non-destructive testing, offering broad application prospects in industries such as automotive, railways, electronics, and oil and gas. The system described here is well suited for large-scale deployment and scenarios with limited funding. It can be used to detect surface cracks in pipelines, weld defects, and coating corrosion. As the technology continues to develop and improve, this system is expected to be increasingly adopted in various industrial sectors, significantly contributing to enhanced production efficiency and product quality.

Although there are certain approximation errors present during the digital signal demodulation process, precision and error analysis reveals that the impact of these errors on changes in the amplitude and phase is minimal. Such minor effects do not significantly affect the detection system and can be either ignored or compensated for.

An analysis of the entire work reveals that the current limitations of the study lie in the fact that it has only implemented a single eddy current probe for detection and analysis, and there exists software latency. This can result in the loss of some defect information from the test object when the probe moves too quickly and restricts a deeper evaluation of and efficiency in detecting surface defects. Hence, the system developed in this paper can be considered a prototype. Based on this prototype, further research will aim to expand and improve this method by enhancing the hardware and optimizing the coordination timing between the host and the lower machine to increase the sampling frequency, reduce or eliminate software delays, and design an eddy current array detection system based on this prototype to enhance detection efficiency. This will enable a more in-depth evaluation of surface defects on the test object. 

## Figures and Tables

**Figure 1 micromachines-15-00775-f001:**
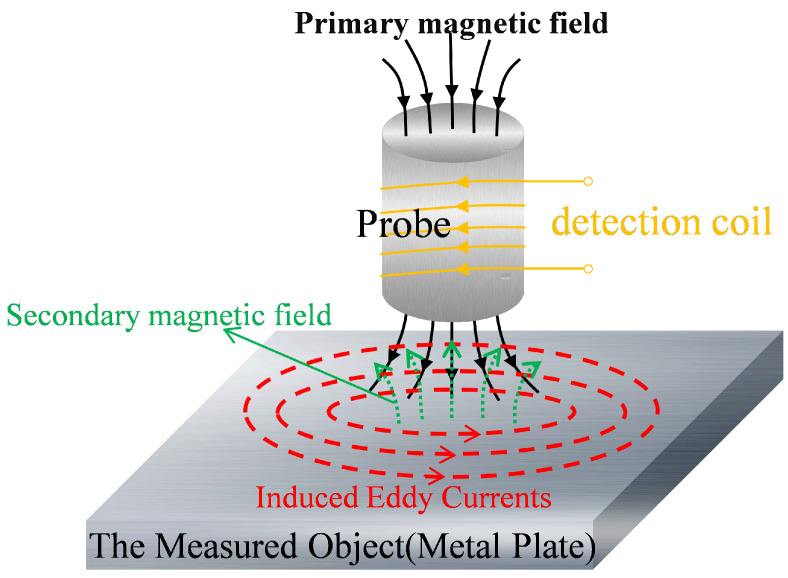
The Eddy Current Model on the test object (interaction between primary and secondary magnetic fields).

**Figure 2 micromachines-15-00775-f002:**
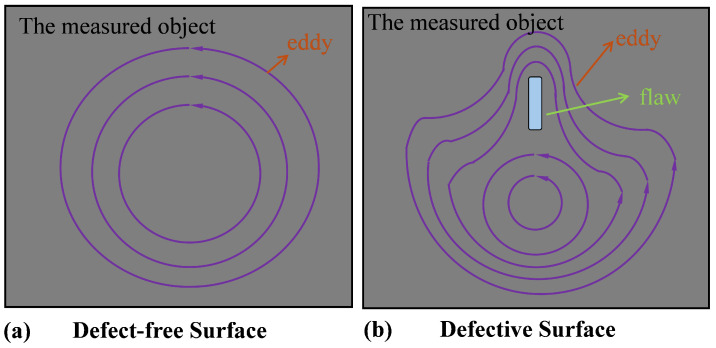
Eddy current effect. (**a**) Eddy current paths when the object’s surface is defect-free. (**b**) Eddy current paths when there is a defect on the object’s surface.

**Figure 3 micromachines-15-00775-f003:**
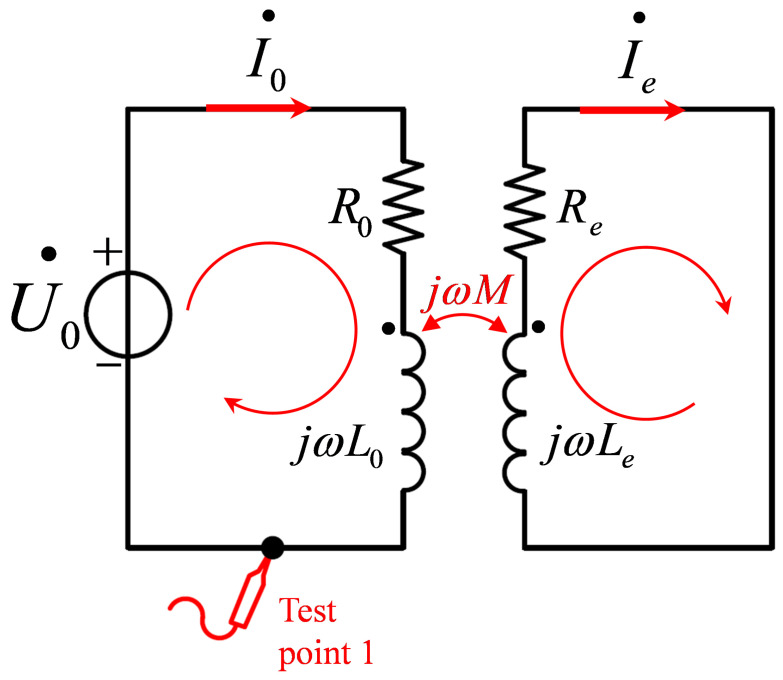
Transformer-Based Mutual Inductance Model between coil and test object.

**Figure 4 micromachines-15-00775-f004:**
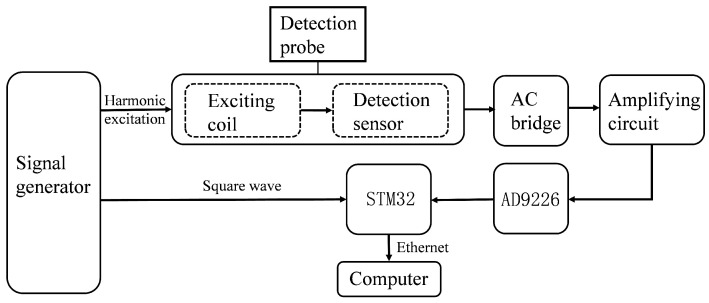
Block diagram of eddy current detection system.

**Figure 5 micromachines-15-00775-f005:**
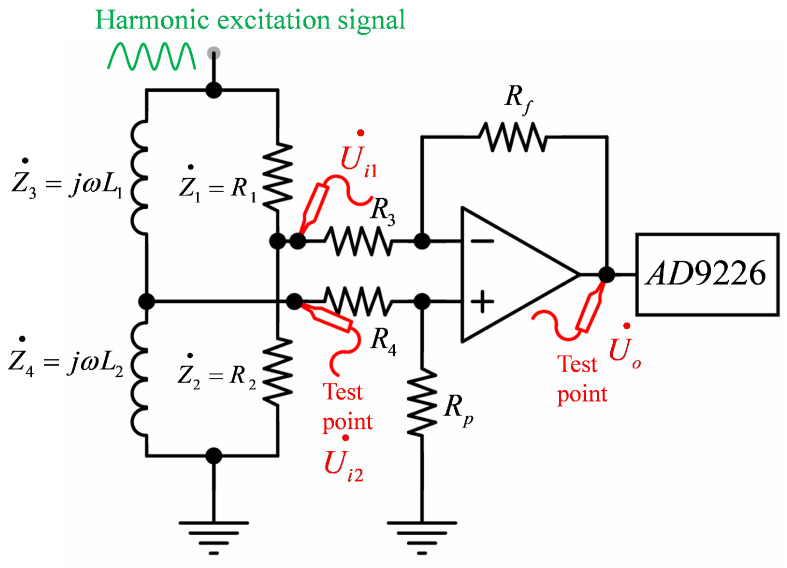
AC bridge circuit.

**Figure 6 micromachines-15-00775-f006:**
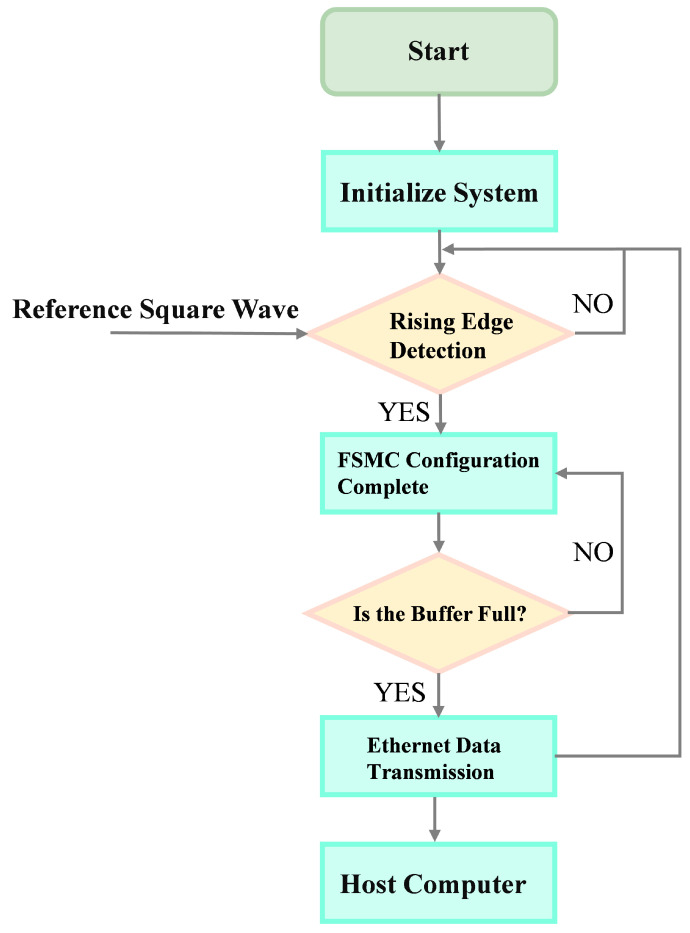
Lower-computer program flowchart.

**Figure 7 micromachines-15-00775-f007:**
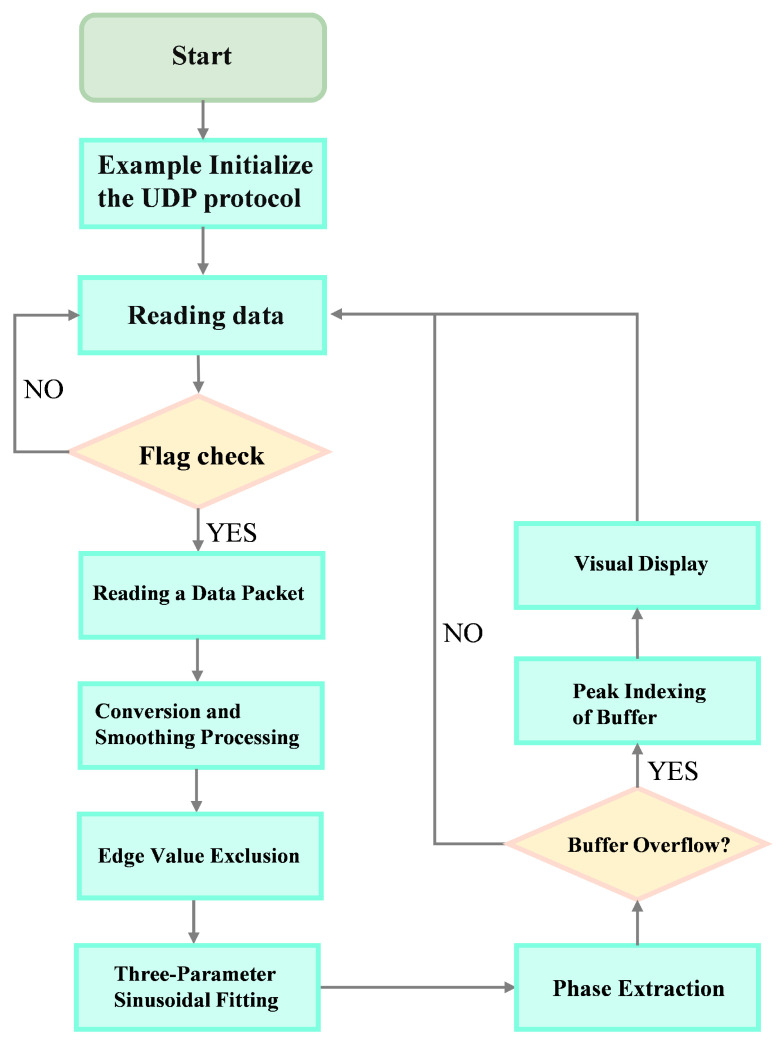
Host computer flowchart.

**Figure 8 micromachines-15-00775-f008:**
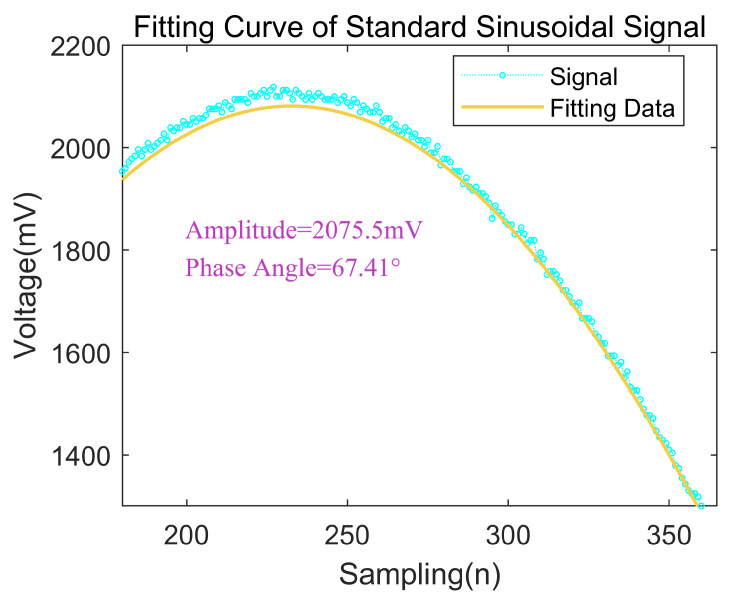
Three-parameter fitting.

**Figure 9 micromachines-15-00775-f009:**
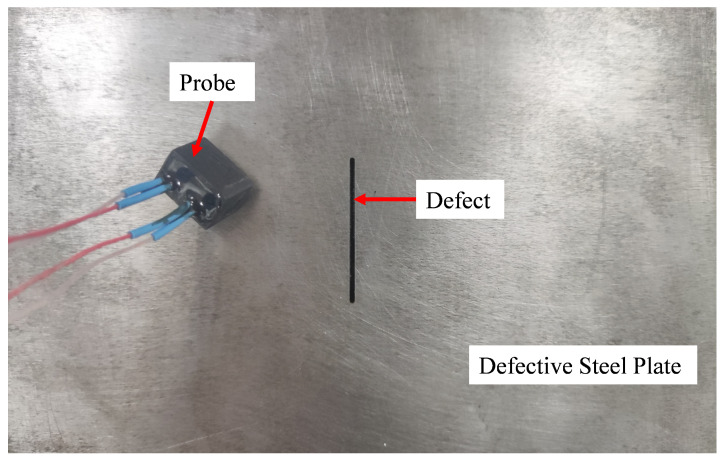
Steel plate under test.

**Figure 10 micromachines-15-00775-f010:**
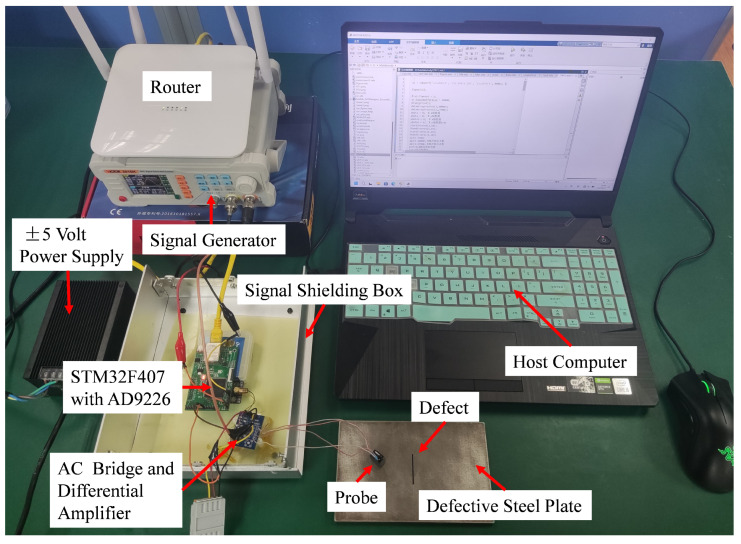
Detecting system.

**Figure 11 micromachines-15-00775-f011:**
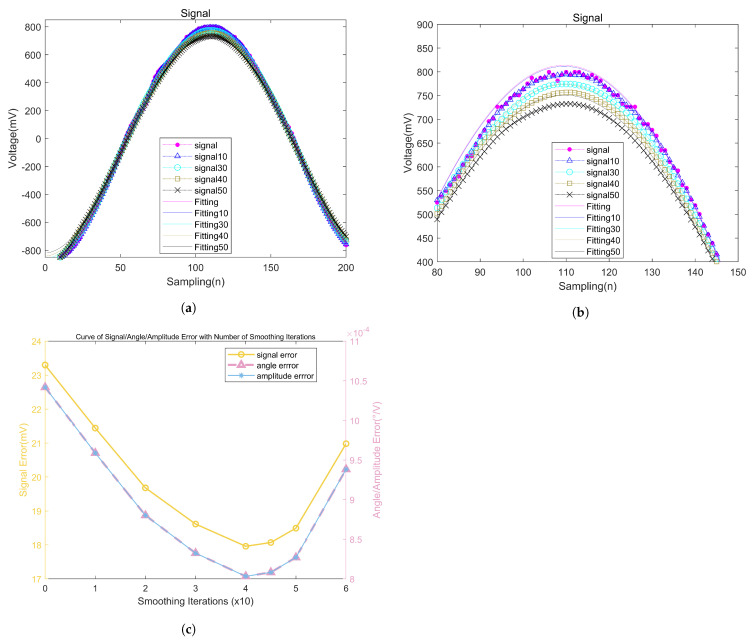
(**a**) Smooth curve and fitted curve. (**b**) Partial enlarged drawing. (**c**) Curve of signal/angle/amplitude error with number of smoothing iterations.

**Figure 12 micromachines-15-00775-f012:**
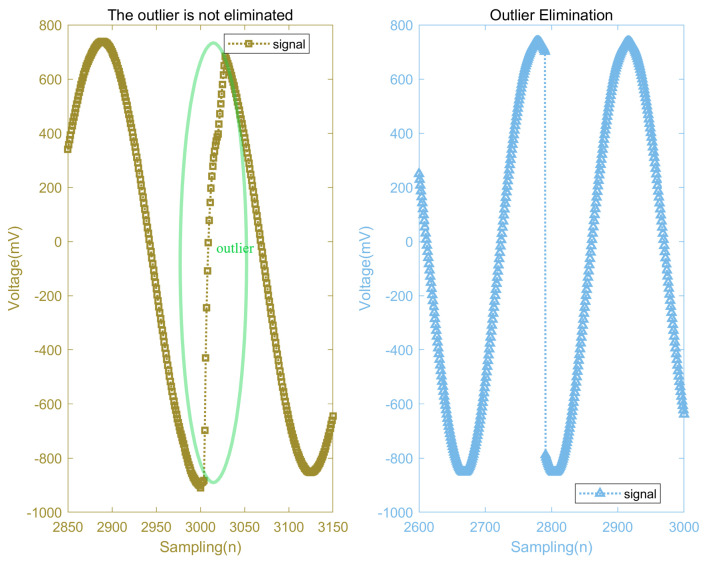
Edge value elimination comparison.

**Figure 13 micromachines-15-00775-f013:**
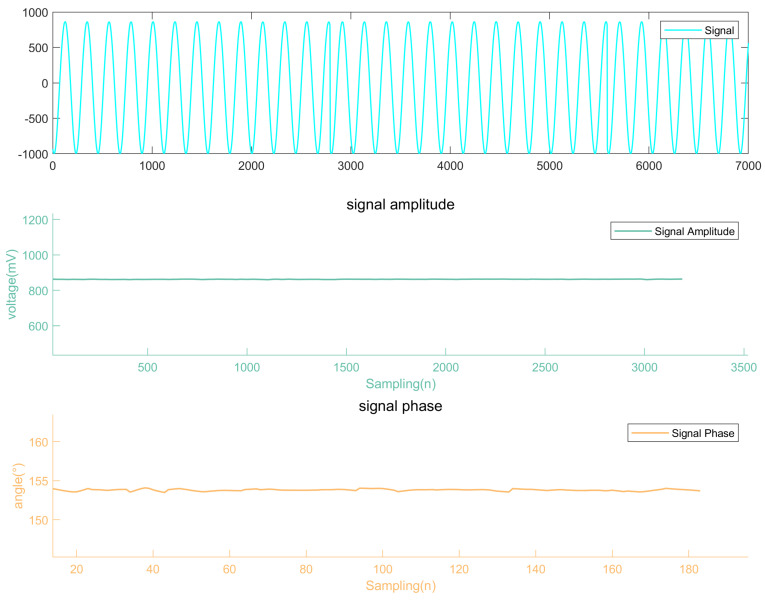
Static measurement of defect-free surface.

**Figure 14 micromachines-15-00775-f014:**
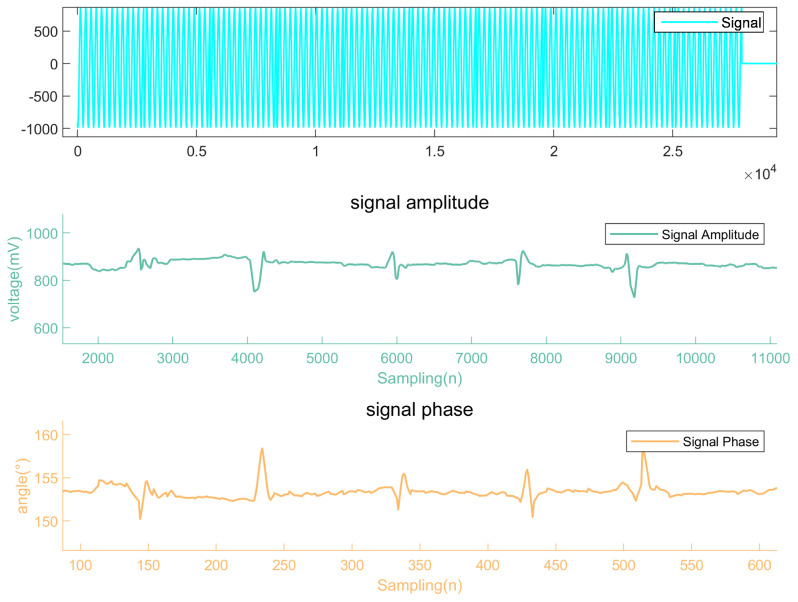
Dynamic measurement of defective surface.

## Data Availability

All data that support the findings of this study are included within the article.

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
