# Peer review of "Digital Eddy Current Detection Method Based on High-Speed Sampling with STM32"

_micromachines, 2024, doi:10.3390/mi15060775_

Round 1

Reviewer 1 Report

Comments and Suggestions for Authors

The paper presents a novel digital eddy current detection method utilizing high-speed sampling with an STM32 microcontroller. The topic is relevant and timely, given the increasing demand for non-destructive testing methods in various industries. The methodology is well-explained, and the results are promising. However, there are several areas that require further clarification and improvement.

1. Include a comparison with recent digital approaches in eddy current testing. Highlighting how the STM32-based method compares in terms of performance, cost, and application scope would provide a clearer picture of the method’s advantages and limitations.

2. The discussion should be improved. Offer a deeper interpretation of the experimental results. Discuss the implications of the findings in a broader context, such as potential industrial applications or scalability issues. Additionally, consider discussing any limitations or challenges encountered during the experiments.

3. Why five defect signals were observed, while only one crack was detected.

Author Response

We appreciate all the comments on the manuscript. We have revised the manuscript and the added and modified words are in red color. Our response is enclosed. Please see the attachment.

Reviewer 2 Report

Comments and Suggestions for Authors

The paper under review presents a truly innovative approach to the design and implementation of an eddy-current system for the investigation of electrically conductive materials. The authors have not only proposed but also demonstrated a Digital Eddy Current Detection Method Based on High-Speed Sampling with STM32, a method that offers a more cost-effective alternative to traditional systems. Their work is commendable, especially considering the rarity of measurements up to the frequency of 8.84MHz in common ECT practice. 

The authors' greatest contribution is creating a given system for inspection and its good functionality in the time domain. It is a prototype that has the potential for meaningful use in research. 

Upon careful examination of the article, I have identified several areas that could benefit from improvement. These include inaccuracies and incorrect formulations that, if addressed, could significantly enhance the professional level of the paper. I believe that by providing this feedback, I am offering valuable guidance and support to the authors, which could ultimately lead to a higher quality article.

Here is a list of my comments on the article:

In section 2, the authors present the principle of the ECT method from a circuit theory perspective. However, there are major things that could be improved in this section. It is necessary to follow the labelling of quantities in the usual way /variables, constants, phasors, etc./ In equation 1, the relationship for the phase of the harmonic signal is given. The representation must be clarified whether this is a stand-alone relationship or a supplementary condition to the previous equation. The title of Figure 1 is misleading. The coil representation in this figure could be clearer. Due care must be taken here to plot the situation correctly, with all quantities correctly labelled. The terminology needs to be unified throughout the paper: no flaw is mentioned once, and defect-free the next time. Figure 3: The title of the figure is incorrect. At the same time, the whole figure needs to be redrawn here: the schematic representation of the mutual inductive coupling should be between the coils, not above the circuit. Correct symbols should denote quantities /voltage phasors and current phasors/. If the authors give equations /4,5/ in the article, all quantities that appear in them should be given with an explanation for the reader. In Figure 4, the sine wave must be changed to harmonic excitation. By square wave, the authors assume a logic signal. In equation 6, the impedances need to be adjusted to phasors again. 

In Figure 5, nodes are missing in the wiring diagram. For resistors, the symbol used is inconsistent with the current labelling using standards. In equation 7, the complex number notation regarding impedances must be included. Figure 6: Some labels occur outside the plotted areas, detto Figure 7. Equation 16: What does the special symbol for multiplication shown mean?Fig.8 is very difficult to read in the given colour configuration. Figures 9 and 10 give a very unprofessional impression. I recommend the authors take photographs that are of a higher quality /e.g. in Fig.9, it is clearly visible that the surface of the material is inhomogeneously smeared with paint/varnish/. Of course, its variable thickness also affects the resulting ECT signal.../Fig.11c is very difficult to read, detto Fig.12, 13 and 14. The conclusions in the paper need to be sufficiently supported by the results obtained. 

After studying the article, I have concluded that it contains gross errors and formulations that sharply reduce its professional level. Based on the above, I do not recommend the article for acceptance in its present form.

Author Response

We appreciate all the comments on the manuscript. We have revised the manuscript and the added and modified words are in red color. Our response is enclosed. Please see the attachment. (We must point out that due to compression issues with Word documents, the images provided in the response letter may not have high clarity. However, this issue does not exist in the revised manuscript, where the images are clear. You can refer to the revised manuscript we have provided.)

Round 2

Reviewer 2 Report

Comments and Suggestions for Authors

After incorporating comments, the authors submit a corrected version of the article for review. After reviewing the article, I conclude that the article has a much higher narrative value and logical structure. It is, therefore, more engaging for the reader. The authors have incorporated all the comments and made the necessary technical corrections. There is one more discrepancy in the article after the correction: in Figure 3, the symbol w needs to be corrected to the symbol omega (from the Greek alphabet). Following this correction, I recommend the article for publication.

Author Response

We appreciate all the comments on the manuscript. We have revised the manuscript and the added and modified words are in red color.We have responded to each of the reviewer's comments and made revisions to the manuscript accordingly.Please see the attachment.
